# Benefits of Enacting and Observing Gestures on Foreign Language Vocabulary Learning: A Systematic Review and Meta-Analysis

**DOI:** 10.3390/bs13110920

**Published:** 2023-11-10

**Authors:** Luca Oppici, Brian Mathias, Susanne Narciss, Antje Proske

**Affiliations:** 1The Department of Teacher Education and Outdoor Studies, Norwegian School of Sport Sciences, 0863 Oslo, Norway; 2School of Psychology, University of Aberdeen, Aberdeen AB24 3FX, Scotland, UK; brian.mathias@abdn.ac.uk; 3Psychology of Learning and Instruction, Department of Psychology, School of Science, Technische Universität Dresden, 01062 Dresden, Germany; susanne.narciss@tu-dresden.de (S.N.); antje.proske@tu-dresden.de (A.P.); 4Centre for Tactile Internet with Human-in-the-Loop (CeTI), Technische Universität Dresden, 01062 Dresden, Germany

**Keywords:** gesture, language learning, embodied learning, sensorimotor simulation

## Abstract

The integration of physical movements, such as gestures, into learning holds potential for enhancing foreign language (L2) education. Uncovering whether actively performing gestures during L2 learning is more, or equally, effective compared to simply observing such movements is central to deepening our understanding of the efficacy of movement-based learning strategies. Here, we present a meta-analysis of seven studies containing 309 participants that compares the effects of gesture self-enactment and observation on L2 vocabulary learning. The results showed that gesture observation was just as effective for L2 learning as gesture enactment, based on free recall, cued L2 recognition, and cued native language recognition performance, with a large dispersion of true effect across studies. Gesture observation may be sufficient for inducing embodied L2 learning benefits, in support of theories positing shared mechanisms underlying enactment and observation. Future studies should examine the effects of gesture-based learning over longer time periods with larger sample sizes and more diverse word classes.

## 1. Introduction

### 1.1. Enacting Gestures Enhances Foreign Language (L2) Learning

There is growing consensus that the integration of physical movements such as gestures into the learning process can enhance learning outcomes. This practice has been described using various terms, from body-based learning, to embodied learning, to sensorimotor enrichment [1,2,3,4,5]. Gestures are typically defined as spontaneous hand movements that accompany speech [6]. However, their use extends to various nonverbal scenarios, such as conveying conceptual information without words [7]. Learning interventions that involve gestures have been shown to bolster subsequent performance in a diverse range of tasks. These include, but are not limited to, tasks of spatial reasoning [8], reading comprehension [9], mathematics and numerical cognition [10], visual memory [11], and conceptual learning [12]. 

Interventions centered on gesture-based learning appear especially promising for enhancing foreign language (L2) vocabulary acquisition [13,14,15]. L2 learning is time-consuming and effortful, and L2 vocabulary is typically taught using more passive learning strategies such as memorizing written word lists [16]. Gestures are known to already play an integral role in native language (L1) learning [17], and the meaning of new words can be easily mapped onto symbolic or iconic body movements [18]. Executing gestures during L2 learning may aid learners in mapping L2 vocabulary onto existing semantic representations associated with L1 words [19,20]. 

The production of gestures during L2 encoding has been found in many studies to facilitate the learning of L2 vocabulary [21]. Participants in these studies typically enact gestures that are semantically related to a concurrently presented written or spoken word, such as bringing one’s hand in a “c” shape towards the mouth while hearing the word *drink* in L1 and L2 [22,23,24]. Memory performance improvements have been observed following gesture-based learning relative to auditory-only learning on both free recall [23,25] and cued recall tests [26]. Free recall refers to the recollection of an L1 word and its L2 translation in the absence of any written or spoken cue, and cued recall here refers to the translation of a written or spoken L1 word into its L2 counterpart and vice versa. 

Why does the enactment of gestures serve as such an effective embodied learning strategy? Current theories propose that executing gestures during learning establishes multiple routes to successful memory retrieval following learning. According to the motor trace theory [27], the enactment of gestures during learning generates a motor memory trace that, along with visual and semantic traces, becomes a part of a learned mental representation. This multimodal representation is assumed to be retrieved faster and more accurately than unimodal representations that do not contain motor traces. Motor trace theory bears similarity to reactivation theories of memories for actions [28], which propose that motor regions of the brain that are involved in gesture-based learning become reactivated when the learning content is recalled following learning. Current evidence suggests that motor areas of the cerebral cortex are not only reactivated during the translation of auditorily-presented L2 words that have previously been learned by executing gestures [26,29,30], but that these motor regions functionally contribute to behavioral benefits of gesture-based learning [31]. 

Theories of grounded cognition propose that gesture-based conceptual learning can result in the simulation of sensory input and associated movements that occurred during learning if the concept is subsequently encountered [32], and that this simulation process supports learning outcomes [2]. For example, learning a novel word while performing a gesture during learning, and then hearing that word in a subsequent memory test, might trigger a reconstruction or simulation of the gesture that was performed during learning. This assumption is consistent with several other theoretical accounts such as those emphasizing a role of mental imagery of sensorimotor representations [33] and the predictive coding of sensorimotor information during post-learning unisensory perception (the multisensory predictive coding framework [4]). The simulation of motoric aspects of a memory representation during memory recall or recognition, as well as the reactivation of motor brain regions, has also been referred to as motor resonance [34].

Interestingly, gestures need not always relate directly to word meanings to aid L2 learning. For instance, beat gestures—rhythmic hand or arm movements that emphasize specific words or mark rhythm without conveying specific meaning—have been found to benefit semantic learning [35]. Similarly, pitch gestures, which mirror speech intonation or melody, aid learners in discerning lexical tones and remembering the meanings of L2 words that contain those tones [14]. These findings hint at a relationship between auditory perceptual learning and semantic acquisition in which enhanced recognition of L2 perceptual patterns corresponds to improved L2 vocabulary learning. This is particularly evident in the case of non-tone-language speakers’ learning of tone languages, in which a change in pitch contour can alter word meaning. In such contexts, semantic proficiency has been shown to depend on the ability of a learner to differentiate pitch contours of lexical tones [36].

### 1.2. Observing Gestures Enhances Learning

In most studies on gesture-based learning, learners are asked to mimic gestures that are performed by an instructor, teacher, or actor shown in video recordings (e.g., [13]). Thus, learners in these studies not only *enact* gestures, but also *observe* gestures. Gesture observation can be considered a confounding variable in that it cannot be concluded whether improvements in learning for gesture conditions are caused by executing gestures or simply by viewing gestures, or some combination of these two tasks. Some studies suggest that just observing gestures without performing them can benefit learning and memory. For example, schoolchildren who observed meaningful gestures while learning about principles of geometry learned more than children who learned without gestures [37]. Similar findings have been reported for the learning of mathematical equivalences [38], the comprehension of verbal stories [39], and the learning of concepts from video-recorded lessons [40]. Hald and colleagues [41] also demonstrated that dynamic gesture videos benefit vocabulary learning more than static pictures of the same gestures.

Other studies have suggested that observing another person enact gestures may be just as effective for learning as enacting gestures oneself. In one study, learners memorized word lists by either enacting gestures or by observing an experimenter enact the same gestures [42]. No differences in memory were found between enactment and viewing conditions, suggesting similar effects of both strategies on learning. In another study, participants learned short phrases such as “peel a potato” by reading while viewing a congruent gesture, reading while imitating a congruent gesture, or reading alone [43]. Both the gesture observation and gesture enactment conditions enhanced memory for the phrases relative to the reading-only condition, but memory scores associated with the two gesture conditions did not differ. Finally, a recent meta-analysis on the role of gestures in the comprehension of accompanying speech found that both gesture enactment and observation benefitted speech comprehension compared with speech-only conditions [44]. 

Why might observing gestures serve as an effective learning strategy, and potentially be just as effective for learning as gesture enactment? One possibility, according to some embodiment theories, is that both viewing and executing gestures during learning could trigger similar or identical sensorimotor simulations during subsequent memory recall. For example, the theory of event coding (TEC) proposes that sensory outcomes associated with viewed and executed body movements share a common representational coding in the mind [45], i.e., seeing someone else execute a movement activates one’s own mental representation of that movement. This idea stems from the fact that movement requires contributions of not only the body’s motor system, but also perceptual systems processing sensory feedback [46]. 

Jeannerod’s [47] motor simulation theory provided a more granular account of the mechanisms involved in the TEC by specifying the neural underpinnings of motor simulation. Essentially, the neural processes that take place during action execution may also take place during action observation, but without leading to overt physical movements. In support of this view, many neuroscience studies have found that brain networks for perception and action are highly integrated; the perception and execution of physical movements relies on an overlapping premotor, parietal, and somatosensory network of brain regions [48]. Neural pattern classifiers can identify whether an individual has performed an action or not based solely on the brain’s responses to sensory consequences of the action [49]. The discovery of mirror neurons in macaque monkeys and potentially in humans added another dimension to this debate [50]. Mirror neurons fire both when performing an action and when viewing the same action and are currently thought to contribute primarily to the low-level processing of observed actions such as distinguishing types of grip [51]. If perception and action are tightly coupled, one might predict similar benefits of gesture enactment and observation, because both forms of learning would likely trigger sensorimotor simulations during subsequent recall.

### 1.3. Does Gesture Enactment Benefit L2 Learning More than Gesture Observation?

The evidence on whether enacting gestures during L2 learning is more beneficial than merely observing gestures is mixed. Some studies report greater benefits of gesture execution than observation [52,53], or greater benefits of gesture execution compared to other visual-only strategies such as viewing of semantically related pictures [24,26,54]. These findings favor a superior role of action experience in driving embodiment benefits. On the other hand, a handful of studies have reported no differences between gesture-based and visual-only L2 learning [55,56,57,58]. Andrä and colleagues [13] found that performing gestures and viewing pictures equivalently benefitted eight-year-olds’ learning of novel L2 vocabulary, in favor of similar effects of gesture observation and execution on L2 word recognition memory. Interestingly, Repetto et al. [59] recently found that the enactment of gestures during learning resulted in enhanced post-learning peripheral forearm muscle activity recorded with electromyography relative to viewing gestures during learning, although behavioral measures following gesture enactment and observation did not differ. This finding suggests that physiological measures may be able to detect differences between enactment and observation when such differences are not captured by behavioral outcomes.

Self-generated actions may give rise to more vivid or detailed sensorimotor simulations than memory for actions performed by others, a view taken by the gesture as simulated action (GSA) framework [60]. To demonstrate this possibility, Hostetter and Alibali [61] asked participants to describe memories in two conditions. In one condition, participants described an action that they had previously executed. In another condition, participants described an action depicted by animated characters in a video that they had not previously executed. The participants gestured about three times more when describing the previously executed action compared to the action that had only been viewed. Also consistent with this view are reports that enacting gestures also enhances performance on mental rotation tasks more than observing an experimenter enact gestures [62,63], and neuroscience work showing that the magnitude of motor brain responses during gesture perception is influenced by whether those gestures were previously enacted or only viewed [64]. Sensorimotor simulations during gesture observation may additionally depend on other factors such as the observer’s individual motor expertise and their personal beliefs about an action’s consequences [65,66]. Taken together, this evidence would suggest that action experience, more than observation, benefits learning outcomes.

Benefits of direct action experience over mere observation in learning and memory are predicted by several other theoretical frameworks besides GSA. For example, cognitive load may be reduced by performing actions oneself, i.e., more cognitive resources might be involved in comprehending and internalizing gesture-related material when the gesture is performed by someone else [67,68,69]. Another explanation comes from constructivist or active theories of learning. Rooted in the philosophies of Popper and Piaget [70], these theories emphasize the active role of learners in building knowledge schemas; performing a gesture, as opposed to passively viewing it, might help learners construct a more detailed and structured understanding of L2 vocabulary. Finally, self-referential theories of memory encoding argue that information related to oneself is better remembered than information related to others [71]. By performing a gesture, rather than viewing it, learners might process L2 words more in relation to themselves, leading to enhanced L2 retention. 

Beneficial effects of gesture enactment and observation in the domain of L2 learning may also be influenced by the types of words that are learned [13]. For example, executing or observing gestures related to concrete nouns such as *ball* and *violin* may be more beneficial than executing or observing gestures paired with abstract nouns such as *patience* and *arrogance*, because *ball* and *violin* refer to concepts that are highly tangible and can therefore be efficiently conveyed using gestures [72]. Abstract nouns such as *patience* and *arrogance* may be conveyed by gestures that are less familiar, i.e., more distant from a learner’s prior sensorimotor experience. Benefits of enacting and observing gestures might therefore diverge for abstract words, with gesture enactment being more beneficial than gesture observation. 

### 1.4. The Current Systematic Review and Meta-Analysis

Whether the execution of physical movements during learning is more (or equally) beneficial than observation of physical movements is a key question for advancing our understanding of body-based approaches to education. If movement execution is more effective than movement observation, this would suggest that action experience is a critical component of embodied learning, consistent with simulation frameworks in which motor simulations are strongest for movements that have previously been self-performed [60]. On the other hand, if action observation is just as effective for learning as action execution, this would suggest that action simulation during learning is itself sufficient for inducing embodied learning benefits, consistent with frameworks proposing a shared representational format of percepts and actions [45,73]. The question of how movement execution relates to movement observation is also important given its practical implications for teaching. The enactment of gestures, for example, may be more challenging for educators to integrate into pedagogy than gesture observation. 

We conducted a systematic literature search and meta-analysis comparing benefits of the enactment and observation of gestures in the domain of L2 vocabulary learning. L2 vocabulary learning provides a fruitful testing ground for comparing effects of enactment and observation, as relatively few studies in other domains directly contrast these two learning techniques while also using similar stimuli across studies. Despite the controversy regarding the roles of observed versus enacted gestures in benefitting learning, no prior meta-analyses have compared effects of observing and enacting gestures on L2 learning outcomes. Only studies in which individuals mimicked gestures performed by an instructor or actor and in another condition (or group) simply observed the instructor’s gestures were included in the meta-analysis. The study designs were limited to experimental, quasi-experimental, and crossover designs that included a control condition to allow for a causal interpretation of the interventions. We had three main aims. First, we aimed to test whether gesture enactment is a more effective or equivalently effective learning strategy than gesture observation, as well as the size of the overall effect. Second, we aimed to examine effects of gesture enactment and observation on several L2 vocabulary learning outcomes including both free and cued recall vocabulary tests. Third, we aimed to assess whether benefits of gesture enactment and observation might be influenced by word concreteness and word type, the number of word repetitions during learning, and the overall number of L2 words presented. 

## 2. Methods

The guidelines proposed by the Preferred Reporting Items for Systematic Reviews and Meta-Analyses (PRISMA 2020; [74]) were followed. The methods of this systematic review were registered post data collection at the Open Science Framework (https://doi.org/10.17605/OSF.IO/JADB3 (accessed on 19 October 2023;)).

### 2.1. Eligibility Criteria

The study inclusion criteria were set using a PICOS statement [75]:P(population): human participants (both with and without learning disability);I(intervention): learning second-language vocabulary with the aid of gesture enactment;C(comparator): learning second-language vocabulary with the aid of gesture observation;O(outcome): Outcome related to memory (recall test) and use of the new vocabulary;S(study design): Experimental, quasi-experimental, and crossover study designs with a control condition.

Furthermore, only peer-reviewed studies published in the English language were included.

### 2.2. Information Sources and Search Strategy

Three databases were searched: Scopus, Web of Science, and PsycINFO. The search was performed on 17 November 2021, and it was updated on 12 April 2023. Additionally, the references of the studies included in the review and relevant review articles were screened as a means of identifying additional relevant studies.

The search string comprised the following syntax: (L2 OR “second language” OR “foreign language” OR foreign OR novel OR novel-word) AND (vocabulary OR word) AND (learning OR acquisition OR recall OR retrieval OR encoding OR memory OR teaching) AND (gesture OR “hand movement” OR embod* OR enact*).

### 2.3. Selection Process

The records identified through database searching were exported into Endnote X9 software (Clarivate, Philadelphia, PA, USA); duplicates were first removed automatically, and then manually checked for unrecognized duplicates. The screening was first performed on manuscript titles, then on abstracts, and ultimately on full article texts. Two authors (L.O. and A.P.) independently screened the records, cross-checked their results, and resolved any conflict. If consensus was not reached, then a third author (B.M.) was consulted.

### 2.4. Data Collection Process and Data Items

The following data were extracted from each study: general study information (author, year, study design), sample characteristics (size, native and L2 languages), material and stimuli (word list, volume of practice and procedure), intervention, assessment tasks and outcome measures. One author (L.O.) extracted the data, and a second author (A.P.) assessed data accuracy.

### 2.5. Study Risk of Bias Assessment

Risk of bias tools developed by the Cochrane group were used for the risk of bias assessment [76]. The RoB 2 tool was used for assessing randomized trials and the RoB 2 for crossover trials was used for assessing studies with a crossover design (https://www.riskofbias.info/welcome/rob-2-0-tool, accessed on 1 March 2022). RoB 2 comprises five bias domains: (i) randomization process, (ii) deviations from the intended interventions, (iii) missing outcome data, (iv) measurement of the outcome, and (v) selection of the reported results. RoB 2 for crossover trials contains these five domains and an additional domain for bias arising from period and carryover effects. Signaling questions help the assessment of the potential bias in each domain. For example, “Was the allocation sequence random?” and “Was the allocation sequence concealed until participants were enrolled and assigned to interventions?” are signaling questions for the randomization domain. Each domain has three possible risk assessment outcomes: low risk, some concerns, and high risk. An overall outcome, corresponding to the highest risk across domains was calculated for each study (i.e., if the risk was *some concerns* in one domain only but *low* in all other domains, the overall risk was *some concerns*). The results of the risk of bias assessment are presented using the traffic light system: green (low), yellow (some concerns), and red (high), created with the Risk-of-bias VISualization (robvis) tool [77]. Two authors (L.O. and A.P.) independently assessed the risk of bias using the Excel spreadsheets available at https://www.riskofbias.info/welcome (accessed on 1 March 2022); results were cross-checked and any conflicts were resolved.

### 2.6. Synthesis Methods

A meta-analytic integration of the results was performed to synthesize the existing literature on the examined research question using the Comprehensive Meta-Analysis software, version 3 (Biostat Inc., Englewood, CO, USA). Standardized mean differences (Hedge’s *g* [SMD]) and 95% confidence intervals (CI) were computed between the “observe” and “perform” learning interventions based on means and *SD*s in recall tasks, and the number of participants. These data were directly extracted from articles, and, when data were not publicly available, the corresponding author was contacted. The correlation between paired conditions was not provided for any crossover trial studies and was therefore assumed to be 0.5 [78]. When a study provided outcomes at different time points (e.g., post-test and retention), means and *SD*s were averaged across the time points. The initial analysis aggregated outcome measures in each study and then a subgroup analysis was performed on the individual outcomes (i.e., free recall, L1–L2 and L2–L1 cued recall). A subgroup analysis was also performed on pitch and iconic gestures. Further, meta-regression was performed with Comprehensive Meta-Analysis software to evaluate how potential covariates interacted with effect sizes: (i) concreteness of words (concrete, abstract, and a mix of the two), (ii) type of words (nouns and verbs), (iii) volume (number of repetitions during practice), and (iv) number of words to be learned. Due to low study sample sizes, each covariate was added to the meta-regression model individually. A random effect model was chosen for all analyses to account for the diversity in sampling in the included studies [79]. The *SMD*s of individual studies and across studies (i.e., the mean effect) were presented using a forest plot. *SMD*s of 0.2, 0.5, and 0.8 were interpreted as small, moderate, and large effect sizes, respectively [80]. The statistical significance threshold was set to alpha = 0.05. 

Heterogeneity (i.e., the dispersion of the effect sizes across studies) was evaluated using the prediction interval [81,82,83], calculated using CMA prediction interval software (Biostat Inc., Englewood, USA; available at https://www.meta-analysis-workshops.com/, accessed on 1 March 2022) [84]. Importantly, the CI of the *SMD* represents the precision of the estimate of the mean, while the prediction interval represents how disperse the true effect sizes are across the populations. I^2^ (percentage of variance in observed effects that reflect variance in true effects rather than sampling error), tau^2^ (variance of true effects), and tau (standard deviation of true effects) were also reported for completeness. Lastly, potential publication bias was assessed using the trim-and-fill method on funnel plot asymmetry [85].

### 2.7. Quality of the Evidence Assessment

The level of evidence of the outcomes of interest was assessed using the Grading of Recommendations Assessment, Development, and Evaluation (GRADE) approach [86]. Each outcome was evaluated individually, and the level of evidence was rated as high, moderate, low, or very low based on nine assessment criteria. The study design serves as the starting point: RCTs are rated high and non-RCTs are rated low. Then, the initial rating can be downgraded based on risk of bias, inconsistency, indirectness, imprecision, and publication bias, and/or upgraded based on the magnitude of the effect size, dose response, and no plausible confounding [87]. Two authors (L.O. and A.P.) independently graded the level of evidence independently, cross-checked their results, and resolved any conflicts.

## 3. Results

### 3.1. Search

The search resulted in a total of 1298 studies (531 in Scopus, 511 in Web of Science, and 256 in PsycINFO). After the removal of duplicates, 757 studies were screened and 741 were excluded based on their title or abstract. The full texts of the resultant 16 studies plus one study identified through reference lists were screened, and a total of 7 studies were included in the meta-analysis (Figure 1). Throughout the screening, studies were excluded because either they did not deal with learning, did not examine the learning of L2 vocabulary, or did not compare gesture enactment with gesture observation (these studies compared gesture enactment with control conditions that required viewing pictures or written text). 

### 3.2. Overview of Study Characteristics

A detailed description of the included study characteristics is presented in Table 1. The studies involved a total of 309 participants, with sample sizes ranging from 11 to 88 participants (*M* = 44 participants, *SD* = 27). Six studies recruited adults (mostly university students) and one study recruited children, with ages ranging from 11.5 to 25.5 years (*M* = 20 years, *SD* = 5). Four studies conducted a parallel-randomized controlled trial (i.e., between-participants design), while three studies conducted a crossover-randomized controlled trial (i.e., within-participants design). All studies compared a group or condition in which participants observed and imitated the gestures performed by an instructor (enactment condition) with a condition in which participants simply observed the instructor’s gestures and were instructed to not move (observation condition). The experimental setup and procedure varied across studies: the intervention included iconic gestures in five studies (two studies also included meaningless and incongruent conditions) and pitch gestures in the other two studies; the number of words-to-be-learnt ranged from 8 to 45 (*M* = 25 words, *SD* = 16); the number of training sessions ranged from 1 to 4 (*M* = 3 sessions, *SD* = 1); the total number of repetitions per word ranged from 3 to 28 (*M* = 17 repetitions, *SD* = 9). All studies assessed vocabulary learning using cued recall tests (L1 to L2 translation, L2 to L1 translation, and object identification), and two studies additionally used a free recall test.

### 3.3. Study Risk of Bias

The methodological quality of the studies was determined to be medium to low: two studies presented “some concerns”, and the others a high risk of bias (Figure 2A). The high risk of bias was primarily driven by a lack of reported information regarding assessor blinding in the measurement of the outcome (Figure 2B). This is highly relevant in recall tasks, which are typically subjectively rated. The randomization procedures presented at least “some concerns” in all studies, as these were poorly reported, making it unclear how the participants were randomized into experimental groups or conditions. Furthermore, some studies did not specify whether all participants and/or outcomes were included in the final analysis. 

### 3.4. Synthesis of Results

The synthesis of results is presented in detail in Table 2 and visually presented in Figure 3 and Figure 4. Considering the experimental conditions of the included studies, the synthesis refers to a comparison between observation only (observation condition) and observation plus enactment (enactment condition) of gestures. The standardized mean difference (Hedge’s *g* (SMD)) was trivial to small and nonstatistically significant for the overall effect and all subgroup analyses. The small effects are in favor of the gesture enactment condition in both cued recall tests (L1 to L2 translation and L2 to L1 translation), with the confidence intervals crossing 0 (Figure 4B,C). Furthermore, the prediction interval showed a large spread in all analyses, spanning large effects in favor of both directions (gesture observation and gesture enactment conditions), indicating a large dispersion of the true effect across populations and study designs (Table 2). The subgroup analysis of gesture types did not show differences between pitch and iconic gestures (Table 2), and the meta-regression did not show any statistically significant interaction between any covariate and the effect size (Table 3). Lastly, the funnel plot asymmetry and the trim and fill method did not detect any publication bias (no asymmetry and the SMD did not change). 

### 3.5. Quality of the Evidence

The quality of the evidence was determined to be low for all performance outcomes (free recall, cued L1–L2 translation, and cued L2–L1 translation) and, consequently, also for the overall outcome (see Table 4).

## 4. Discussion

The aim of this systematic review and meta-analysis was to examine whether enacting gestures might serve as a more, or equally, effective strategy for L2 vocabulary learning than merely observing gestures. We evaluated effects of gesture enactment and observation on both free and cued recall vocabulary tests. We also examined potential influences of several study design features such as the number of stimulus repetitions during training on effects of gesture enactment and observation strategies. There were three main findings. First, the meta-analysis uncovered no difference between gesture enactment and observation strategies on an aggregate measure of L2 vocabulary learning, suggesting that both strategies may be equally effective in how they enhance L2 vocabulary learning. Second, despite some individual studies showing gesture enactment to be more effective than observation on cued recall vocabulary tests (e.g., [14]), our meta-analysis revealed no superior benefit of enactment on either cued or free recall of recently learned L2 vocabulary. Third, relative benefits of gesture enactment versus observation were not found to be mediated by the number of L2 words learned, how many times they were presented during L2 training, L2 word concreteness, or L2 word type variables. Taken together, these findings suggest that viewing and enacting gestures during learning may lead to the formation of comparable L2 vocabulary representations, which similarly benefit post-learning L2 memory, consistent with theories proposing a common representational coding of perception and action. Prior experience performing specific gestures may provide a template for the internal sensorimotor simulation of actions that takes place during gesture observation, challenging the notion that execution is a critical component of body-based learning. The current findings, however, should be interpreted with caution given the limited literature on the topic and its methodological shortcomings.

### 4.1. Equivalent Benefits of Gesture Enactment and Observation on L2 Learning

The lack of a consistent benefit of gesture enactment relative to gesture observation on L2 learning in the studies examined here challenges the notion that action execution is a critical component of embodied learning strategies. Equivalent benefits of gesture enactment and observation adjudicate between theories in which enacting, rather than observing, gestures gives rise to more detailed or stronger simulations during subsequent retrieval (e.g., [60]) and theories in which enacted and observed gestures share the same representation [45,73]. The GSA framework predicts greater benefits of gesture enactment than observation on L2 learning, inconsistent with the current meta-analysis results. According to frameworks in which execution and observation are similarly or identically coded, the same neural networks that support gesture observation during L2 learning also support gesture execution. The simulation of these viewed actions during post-learning recall is just as effective as the simulation of actions that were previously executed. Thus, gesture observation, rather than enactment, may be sufficient for inducing embodied learning benefits in the case of L2 vocabulary learning. 

The current meta-analysis suggests that action simulation that takes place during action observation is itself sufficient for inducing embodied learning benefits. In other words, action execution may not, in fact, be a critical aspect of body-based learning, as long as internal action simulations can still take place. Such simulations can take place during gesture observation not only, presumably, due to strong interrelation between perceptual and action representations [45], but also due to participants’ extensive prior experience executing the gestures that they are instructed to observe. The word *drink*, for example, is strongly associated with the action sequence of moving one’s hand toward one’s mouth. Dargue, Sweller [44] demonstrated that observing gestures benefits language comprehension only when the gestures are highly iconic, and therefore likely highly familiar. Gestures associated with concrete words tend to be rated as more iconic than gestures associated with abstract words [24]. The greater iconicity of concrete gestures could explain why gesture observation was found to benefit the learning of concrete L2 words more than the learning of abstract L2 words. Thus, gestures executed at earlier stages of learning—in this case, L1 learning—could bootstrap observational learning by providing an internal model for sensorimotor associations. 

In the studies assessed here, gesture enactment was accompanied by the observation of a model performing the same gestures. This means that observation was consistent across both “enactment” and “observation” scenarios. The absence of a distinct advantage when enacting gestures aligns with the notion that observation alone can give rise to sensorimotor simulations that are just as beneficial as overt action. Further, the current analysis does not indicate that combining enactment with observation *negatively* impacts L2 learning relative to observation alone, as would be predicted by theories emphasizing dual task interference effects in activities that involve physical movements [91]. Therefore, while enactment does not appear to provide an additional advantage relative to mere observation based on the current limited set of data, it does not detract from the learning experience either. This neutral effect of combined enactment and observation could suggest that, when gestures are incorporated into L2 learning, the primary driver of the learning benefit might be the observation and internal simulation, rather than the physical enactment.

### 4.2. Influences of Test Type

Equivalent benefits of gesture enactment and observation were found across multiple language learning outcomes, including free recall, L1–L2 translation, and L2–L1 translation tests. Gesture enactment has previously been shown to enhance performance on both free and cued recall tests [23,25] tests. The current set of results extends these previous findings to suggest that gesture observation can also enhance free and cued recall of L2 words and their L1 translations. In grounded cognition terms, equivalent benefits of gesture enactment and observation across multiple test types suggest that the presence of an external memory cue, i.e., in a cued recall situation, may not be a prerequisite for the simulation of sensory and motor features associated with both viewed and executed gestures. Gesture observation and enactment may benefit the free recall of L2 words, in particular, as free recall tests tend to be more difficult than cued recall tests [92]. It should be noted, however, that there was a small effect in favor of gesture enactment in L1–L2 and L2–L1 cued recall tests that did not reach statistical significance. The confidence interval in both recall tests ranged from a trivial effect in favor of observation to a moderate effect in favor of enactment, suggesting a potential trend of gesture enactment being more beneficial than gesture observation for cued recall memory. This should be further examined in future studies.

### 4.3. Limitations and Recommendations

In addition to the low overall number of studies that could be included, one limitation of the meta-analysis performed here is that none of the studies focused exclusively on abstract words. Of the seven studies included, four investigated effects of gesture enactment and observation on the learning of concrete L2 words, and the other three investigated effects on mixture of concrete and abstract words without separately analyzing effects for each word type. Concrete words refer to objects that can be perceived by the body’s sensory systems [72], and are therefore typically already associated with specific gestures prior to gesture-based L2 training. This is not the case for abstract words, such as *patience*, which cannot be seen or touched. One reason gesture observation was found to be just as helpful for remembering L2 words, as gesture enactment in our meta-analysis could be that observed concrete word gestures are more easily simulated or encoded than observed abstract word gestures. As such, we argue that the results of this meta-analysis primarily generalize to the learning of concrete L2 words. The meta-regression analysis showed a potential (not statistically significant) mediating role of word concreteness and future studies should compare the enactment and observation of abstract word gestures to examine whether the finding that these two learning strategies equivalently benefit L2 learning generalizes to other word types. 

The current meta-analysis revealed a spread of true effects spanning situations under which gesture enactment outperforms gesture observation, and vice versa. This indicates that study- and sample-specific variables may have an influence on the magnitude and direction of potential differences between enactment and observation. However, the meta-regression analysis revealed no influences of several stimulus- and procedure-related factors, such as the number of repetitions of each L2 stimulus during training, on effects of enactment and observation, or interactions between these factors. This could indicate that the influences of such variables were not large enough to be detected in our moderation analyses, or the overall number of studies included in the moderation analyses were too low to provide meaningful moderation estimates. If the former is confirmed by future studies, it will further bolster the conclusion that both gesture enactment and observation strategies promote L2 vocabulary learning. 

The studies included in our meta-analysis have additional limitations. Several studies faced design issues such as shortcomings in the randomization assignment of participants to experimental conditions, a lack of blinding among experimenters, and poor design of outcome assessments (the lack of retention tests). This resulted in the assessment of “some concerns” with two of the included studies, and a high risk of bias in the remaining studies. Although there are some exceptions [13,24,31,93], the vast majority of studies on gesture-based L2 learning have examined only short-term learning using pre- and/or post-tests. This constraint may also have masked potential differences in the effectiveness of enactment and observation learning strategies, as the production of gestures during learning results in less rapid decay of L2 representations compared to visual-only strategies over a six-month period [26]. We recommend that future work compares the time course of effects of gesture enactment and observation over a longer timeframe. 

A pivotal takeaway from our analysis underscores the need for not only more data that compares learning while viewing and enacting gestures, but also more rigorous research methodologies. The methodological limitations of the studies analyzed here impede the formation of a clear consensus and call into question the reliability of existing data. To enhance the quality of future research, it is vital that studies ensure comprehensive randomized assignment of participants to learning conditions or the use of within-group control conditions, utilize blinding techniques, and test large sample sizes when feasible [94,95]. Studies could consider implementing peer reviews at the design stage or preregistering study protocols, pursuing replication to validate findings, and involving multiple research centers to increase sample sizes. To truly understand the effects of gesture-based learning, it will be essential to uphold stringent research practices.

Two additional variables to explore in future work are the degree of congruency between gestures that are performed or observed during L2 learning and the meanings of L2 words, as well as the role of self-generated (i.e., self-created) gestures in L2 learning. The enactment of gestures that are incongruent with word meanings and the enactment of meaningless gestures—movements that do not carry inherent symbolic or semantic value—may be detrimental to L2 learning rather than facilitatory [15]. Manipulations of between-modality congruency may help in quantifying the role of physical activity or engagement in learning, independent of a gesture’s semantic content. On a related note, the role of self-generated gestures in L2 learning remains largely unexplored. Self-generated gestures allow learners to create their own physical representations of L2 vocabulary, potentially leading to a deeper, more personalized understanding. Evidence from other domains suggests that self-generated content can enhance memory retention and understanding [96,97]. It is possible that when learners create their own gestures, they form stronger or more memorable representations of the vocabulary, which could lead to greater benefits of enactment relative to observation alone.

## 5. Conclusions

We conclude that gesture enactment and observation may, under some conditions, equivalently benefit the acquisition of novel L2 vocabulary. Such conditions likely include the learning of novel concrete words whose referents are familiar and strongly associated with specific body movements. The lack of differences between gesture enactment and observation in a variety of outcomes, including free and cued recall tests, suggests that experiencing an action oneself need not always be a critical component of embodied learning. Indeed, viewing others perform familiar actions may be just as effective. The finding that gesture enactment and observation were equivalently beneficial in the experimental studies evaluated here has consequences for the way that classroom teaching practices are designed. Gesture observation may be more easily integrated into pedagogy than enactment. The wide spread of true effects across the included studies suggests that future research examines specific conditions under which observing or executing gestures might be more or less beneficial, such as the learning of abstract L2 words. 

## Figures and Tables

**Figure 1 behavsci-13-00920-f001:**
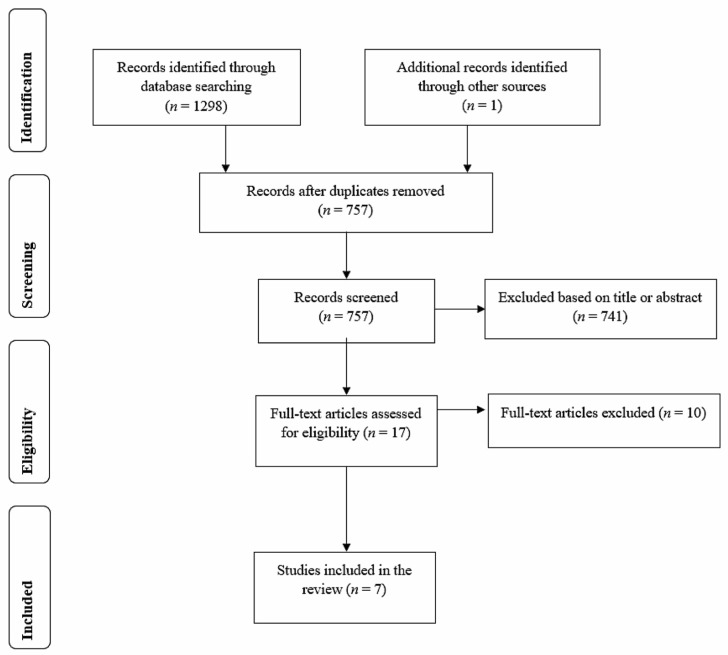
Flow diagram of the meta-analysis search and study selection process.

**Figure 2 behavsci-13-00920-f002:**
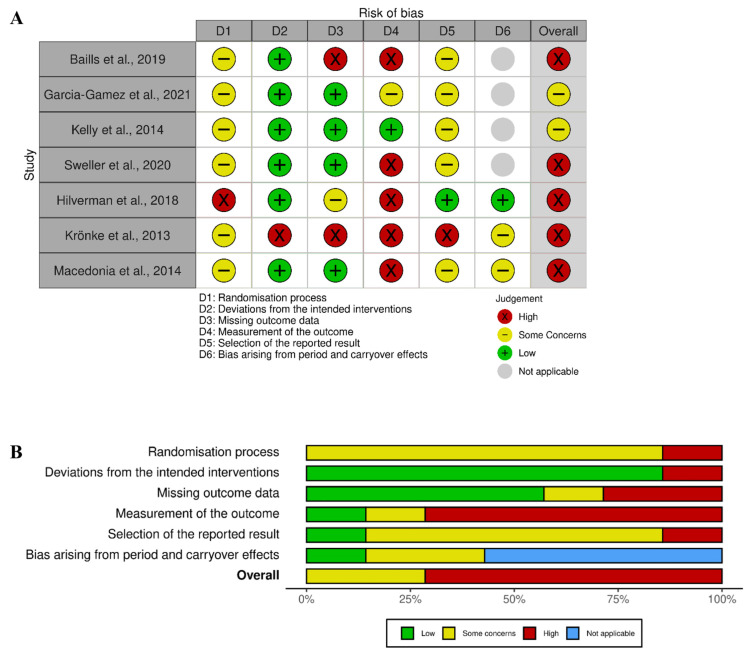
Results of the risk of bias assessment for studies included in the meta-analysis [14,55,56,58,88,89,90], displayed with a traffic light plot for each individual study (panel **A**), and a summary plot (panel **B**). Risk of bias was evaluated with the RoB 2 (first four studies) and RoB 2 for crossover trials (last three studies) tools [76].

**Figure 3 behavsci-13-00920-f003:**
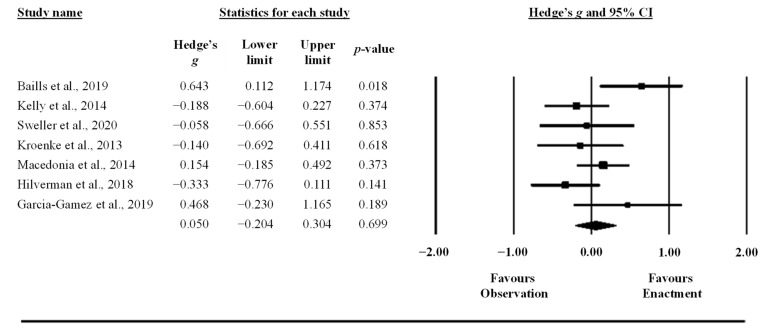
Forest plot showing the differences between observation and enactment of gestures on recall tasks (free and cued recall, and object identification tasks are combined) [14,55,56,58,88,89,90].

**Figure 4 behavsci-13-00920-f004:**
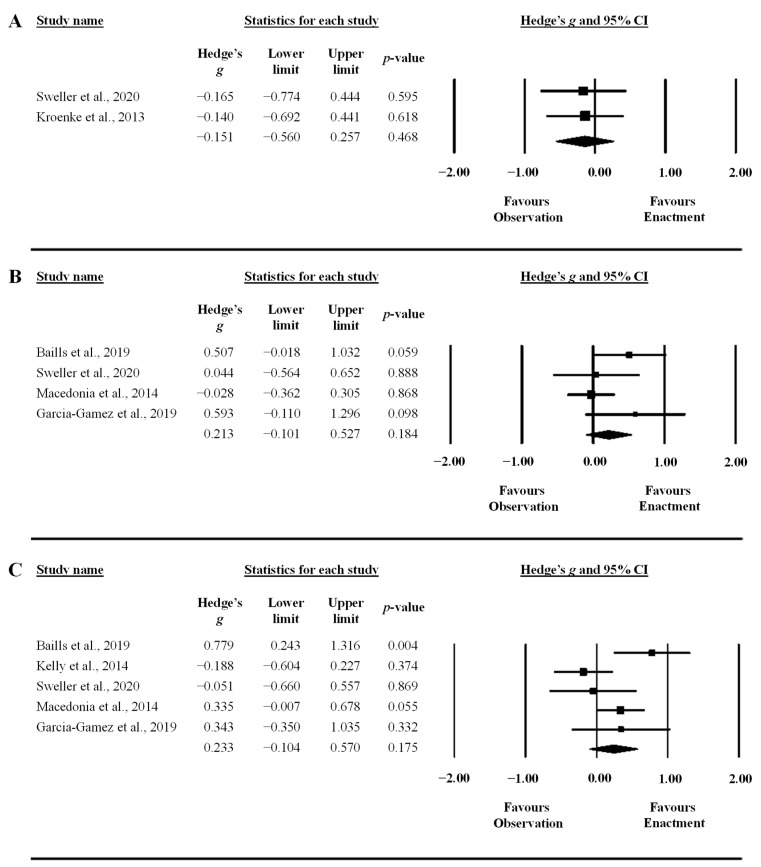
Forest plots showing the differences between observation and enactment of gestures on free recall performance (panel **A**), cued L1–L2 recall (panel **B**), and cued L2–L1 recall (panel **C**) [14,55,56,58,88,89,90].

**Table 1 behavsci-13-00920-t001:** Summary of meta-analysis study characteristics.

Study Information	Sample Characteristics	Intervention	Groups and Type of Gestures	Assessment and Outcome Measures
Baills et al. [14], Parallel-Randomized Controlled Trial (RCT)	*n* = 56, healthy. Age: 20 ± 1.5. Catalan speakers, novices in the target L2 language: Mandarin Chinese. 2 groups: 1 experimental (enactment) and 1 control (observation).	Word list: 12 words (mix of concrete and abstract nouns). Volume: One training session for a total of 3 repetitions per word. Stimuli: Each block contained 2 words paired for their tone similarity: first, a written Chinese word with the Catalan translation, and then a video of an instructor performing a gesture and saying the word.	Both groups viewed videos of an instructor performing *pitch gestures*, i.e., gestures that mimic tone characteristics. Experimental (enactment): Participants enacted the instructor’s gestures and repeated the target word out loud. Control (observation): Participants repeated the target word out loud without any movement.	Word-meaning recall test: Translation of words from Mandarin to Catalan. Word-meaning association test: Participants were presented with a pair of 2 words in Catalan, heard only one of the 2 words in Mandarin, and had to select the correct translation.
García-Gámez et al. [55], Parallel-RCT	*n* = 39, healthy. Age: 21 ± 3. Spanish speakers, novices in an artificial language (Vimmi). 2 groups: 1 experimental (enactment) and 1 control (observation). 4 teaching conditions within each group: no, congruent, incongruent, and meaningless gestures.	Word list: 40 words (mix of concrete and abstract verbs) denoting familiar actions in Vimmi (artificial language). The 40 words were split in 4 groups and randomly, counterbalanced assigned to the 4 teaching conditions. Volume: 3 training sessions on 3 consecutive days for a total of 24 repetitions per word. Stimuli: Spanish–Vimmi pair was written at the bottom of the screen, with a video of an actor performing a gesture twice. Then, participants were instructed to read aloud the words twice.	Both groups viewed videos of an instructor performing *iconic* gestures. Experimental (enactment): Participants read aloud the word pairs in Spanish and Vimmi (L1–FL) and enacted the gestures that were presented on the screen. This was performed twice per word pair. Control (observation): Participants read aloud Spanish–Vimmi word pairs (L1–FL) and were instructed to observe and to imagine themselves mimicking the gestures presented on the screen. 3 types of gestures within each group: *Meaningless* (gestures unrelated to the word without any precise meaning); *congruent* (iconic gestures showing the typical action linked to the word); *incongruent* (iconic gestures associated with an action different from the word to-be-learned).	Cued recall test: L1-to-L2 and L2-to-L1 translation. Participants were presented with the full list of words in L1 or L2 and were instructed to translate them in the other language. Tests performed at the end of each training session.
Hilverman et al. [88], crossover-RCT	*n* = 4, patients with bilateral hippocampal damage (HP). *n* = 4, patients with brain damage *n* = 19, healthy. English speakers, novices in artificially created words for common objects. 3 teaching conditions: no, observing, and enacting gestures.	Word list: 8 concrete nouns in artificially created novel names. Volume: 2 training sessions, 4 words in each session, 2 words learned without gestures and 2 words with gestures. Each word was practiced until proficiency was reached. Stimuli: Picture of an object; then, a video above the picture with the experimenter providing the object name; then, participants had to repeat the name (no gesture) and also repeat the gesture (gesture conditions).	The teaching conditions were organized: In the first session, no gesture and performing gesture conditions; in the second session, no gesture and observing gesture conditions. Experimental (enactment): Participants enacted the experimenter’s *iconic* gestures. Control (observation): Participants observed the experimenter’s *iconic* gestures. Control (no gesture): Participants only repeated the word.	Trials to reach proficiency: Number of practice trials to reach proficiency in labeling the objects in the video. Recall test: An image of an object was presented on a screen and participants had to provide the name. Object identification test: Four objects were presented on a screen, the experimenter produced out loud the name of one object, and participants had to assign the name to the object. Recall and object identification tests were performed 30 min after practice.
Kelly et al. [56], Parallel-RCT	*n* = 88. Age: 18–23. English speakers, novices in Japanese. 4 groups: 2 experimental (enactment) and 2 control (observation).	Word list: 20 Japanese words (mix of concrete and abstract nouns), grouped in 10 pairs that contrasted in length of vowels. Volume: 4 training sessions over 2 days for a total of 12 repetitions per word. Stimuli: (1) video of an actor saying the L2 word and performing the corresponding tone gesture; (2) translation in English; (4) repetition of (3) with participants performing or observing the gesture; (5) repetition of (2). Participants had to stay silent the whole time.	All groups viewed *pitch gestures*. Experimental (enactment): Both syllable and more groups were presented with and enacted gestures. Control (observation): Both syllable and more groups observed gestures.	Cued recall L2–L1 test: Participants were presented with an audio of Japanese words and were instructed to write down the translation in English. Test performed 1–3 day after the last training session.
Krönke et al. [89], crossover-RCT	*n* = 11. Age: 23–28. German speakers, novices in an artificial language. 5 conditions: no, meaningful iconic (enactment, observation), and meaningless grooming (enactment, observation) gestures.	Word list: 42 concrete nouns in an artificial language. Volume: 3 training sessions, for a total of 21 repetitions per word. Stimuli: Written translation in German, then auditory presentation of L2 word, then presentation of gesture (in gesture conditions).	In all conditions, participants verbally repeated the word once per trial. Experimental (enactment): Participants enacted meaningful *iconic gestures* in one condition and *meaningless gestures* in another condition. Control (observation): Participants observed meaningful *iconic gestures* in one condition and *meaningless gestures* in another condition. Control (no gesture): Participants only repeated the L2 word.	Free recall test: Participants were asked to name all word pairs they could remember. Cued recall test: L1-to-L2 and L2-to-L1. Participants were presented with the full list of words in L1 or L2 and were instructed to translate them in the other language. Tests were performed after the first and third training sessions.
Macedonia et al. [90], Crossover-RCT	*n* = 33. Age: 11.5 ± 1. German speakers, novices in an artificial language (Vimmi). 3 conditions: no, observing, and enacting gestures.	Word list: 45 concrete nouns in Vimmi (artificial language). Volume: 4 training sessions for a total of 28 repetitions per word. Stimuli: Written word in Vimmi, translation in German, and a video or static picture of a virtual agent performing an iconic gesture.	The 45 words were divided in 3 blocks of 15 words, and 3 conditions were created for teaching each block. Each participant underwent all 3 blocks. Experimental (enactment): Participants enacted the agent’s *iconic gestures* and repeated the words out loud. Control (observation): Participants observed the agent’s *iconic gestures* without moving and repeated the words out loud. Control (no gesture): Observed a static picture of the agent and repeated the words out loud.	Cued recall tests: L1-to-L2 and L2-to-L1. Participants were presented with the full list of words in L1 or L2 and were instructed to translate them in the other language. Tests performed after each session and one day after the last session.
Sweller et al. [58], Parallel-RCT	*n* = 63. Age: 20.5 ± 3.5. English speakers, novices in Japanese. 3 groups: 1 experimental (enactment) and 2 control (observation and no gesture).	Word list: 10 concrete verbs in Japanese. Volume: 1 training sessions for a total of 6 verbal repetitions, 15 gesture observations, and 12 gesture performance per word. Stimuli: L2 word and L1 translation, then a video of an actor performing a gesture.	Participants in performing and observing groups were presented with *iconic gestures*. Experimental (enactment): Participants enacted the actor’s gestures. Control (observation): Participants simply observed the gestures. Control (no gesture): Participants were not presented with any gesture.	Free recall test: Participants were asked to recall as many L2–L1 word pairs as they could. Cued recall test: L1-to-L2 and L2-to-L1. Participants were presented with the full list of words in L1 or L2 and were instructed to translate them in the other language. Tests performed the same day as training and 1 week after training.

**Table 2 behavsci-13-00920-t002:** Subgroup analysis.

		SMD (95% CI)	*p*-Value	Prediction Interval	*I* ^2^	Tau^2^	Tau
Overall effect		0.050 (−0.20, 0.30)	0.70	−0.63, 0.73	47	0.05	0.23
Subgroup analysis							
Free recall	All gestures (*n* = 2)	−0.15 (−0.56, 0.26)	0.47	NA *	0	0	0
	Pitch gestures (*n* = 0)	NA					
	Iconic gestures (*n* = 2)	Same as “all gestures”					
Cued L1–L2 recall	All gestures (*n* = 4)	0.21 (−0.10, 0.53)	0.18	−0.85, 1.27	33	0.04	0.19
	Pitch gestures (*n* = 1)	NA					
	Iconic gestures (*n* = 3)	0.18 (−0.26, 0.63)	0.42		49	0.08	0.28
Cued L2–L1 recall	All gestures (*n* = 5)	0.23 (−0.10, 0.57)	0.18	−0.83, 1.29	57	0.08	0.28
	Pitch gestures (*n* = 2)	0.28 (−0.68, 1.25)	0.56		87	0.42	0.65
	Iconic gestures (*n* = 3)	0.26 (−0.02, 0.54)	0.07		0	0	0

* The number of studies is too low (<3) to compute the prediction interval.

**Table 3 behavsci-13-00920-t003:** Meta-regression with concreteness, word type, volume of repetitions (volume), and number of words to learn as covariates.

Covariate	Coefficient	Standard Error	95% Lower Limit	95% Upper Limit	Z-Value	2-Sided *p*-Value
**Intercept**	−0.08	0.17	−0.41	0.26	−0.46	0.65
**Concreteness**	0.32	0.27	−0.20	0.86	1.20	0.23
**Intercept**	0.02	0.15	−0.29	0.32	0.11	0.91
**Word type**	0.17	0.34	−0.49	0.82	0.49	0.62
**Intercept**	0.16	0.34	−0.50	0.82	0.47	0.64
**Volume**	−0.01	0.02	−0.04	0.03	−0.34	0.73
**Intercept**	−0.05	0.27	−0.58	0.48	−0.20	0.84
**Number of words**	0.00	0.01	−0.01	0.02	0.46	0.65

**Table 4 behavsci-13-00920-t004:** Assessment of the level of evidence using the GRADE approach. Table adapted from [87]. The three outcomes (free recall, cued L1–L2, and cued L2–L1) are grouped into one table because they had the same rating.

GRADE Criteria	Rating	Comments	Quality of Evidence
*Outcome: Free recall, Cued L1*–*L2, and Cued L2*–*L1*			
Study design	High (RCTs)		Low
Risk of bias	Serious (−1)	High risk of bias	
Inconsistency	No		
Indirectness	No		
Imprecision	Serious (−1)	Wide confidence intervals	
Publication bias	Undetected		
Other (upgrading factors)	No		

## Data Availability

No new data were collected for this analysis.

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
