# Peer review of "Benefits of Enacting and Observing Gestures on Foreign Language Vocabulary Learning: A Systematic Review and Meta-Analysis"

_behavsci, 2023, doi:10.3390/bs13110920_

Round 1
Reviewer 1 Report
Comments and Suggestions for Authors
Please see attached file

Author Response
We thank the Reviewer for their positive and constructive comments that have helped us to improve the manuscript. Please find a point-by-point reply to the comments below; reviewer comments are italicized and author replies are in normal font. Changes in the manuscript are highlighted with red font.
1. Abstract: I think when the authors state than "n = 309" in the metaanalysis, some readers may think they have analyzed 309 different studies, instead of 309 participants. Can that be made clearer, perhaps by saying "we present a meta-analysis of 7 articles and 309 participants..." or something similar?
Response: We thank the reviewer for this helpful comment. We have made the suggested change (Ln 19).
2. Introduction: I wonder why the authors don't at least mention mirror neurons in their discussion of previous research. I know the implications of mirror neurons is tenuous for humans at best, but it gives a nice possible reason that watching gestures may be as effective as actually producing them.
Response: We thank the reviewer for this suggestion. We have added a description of mirror neurons and how they may underpin the benefit of gesture observation to the Introduction of the revised manuscript (Ln 132-145). We now write: “Jeannerod [47] motor simulation theory provided a more granular account of the mechanisms involved in the TEC by specifying the neural underpinnings of motor simulation. Essentially, the neural processes that take place during action execution may also take place during action observation, but without leading to overt physical movements. In support of this view, many neuroscience studies have found that brain networks for perception and action are highly integrated; the perception and execution of physical movements relies on an overlapping premotor, parietal, and somatosensory network of brain regions [48]. Neural pattern classifiers can identify whether an individual has performed an action or not based solely on the brain’s responses to sensory consequences of the action [49]. The discovery of mirror neurons in macaque monkeys and potentially in humans added another dimension to this debate [50]. Mirror neurons fire both when performing an action and when viewing the same action and are currently thought to contribute primarily to the low-level processing of observed actions such as distinguishing types of grip [51].”
3. I think there also needs to be a better argument made for why performing the actual gestures may be important. Under section 1.3, there needs to be a better theoretical explanation of the data.
Response: We have expanded this section to clarify why enacting gestures may be more important for L2 learning (Ln 181-193). We now write: “Benefits of direct action experience over mere observation in learning and memory are predicted by several other theoretical frameworks besides GSA. For ex-ample, cognitive load may be reduced by performing actions oneself, i.e., more cognitive resources might be involved in comprehending and internalizing gesture-related material when the gesture is performed by someone else [67-69]. Another explanation comes from constructivist or active theories of learning. Rooted in the philosophies of Popper and Piaget [70], these theories emphasize the active role of learners in building knowledge schemas; performing a gesture, as opposed to passively viewing it, might help learners construct a more detailed and structured understanding of L2 vocabulary. Finally, self-referential theories of memory encoding argue that information related to oneself is better remembered than information related to others [71]. By per-forming a gesture, rather than viewing it, learners might process L2 words more in relation to themselves, leading to enhanced L2 retention.”
4. Methods: I really like how the authors carefully outline their methodology in this section to control for bias, etc. However, it was really surprising when only 7 articles were included in the actual analysis. This is especially true given that the authors don't provide more detail except that "741 were excluded based on their title or abstract." Can you give more detail about why the other studies were excluded, such as -- did they not specifically compare gesture observation and gesture enactment? Or what were the reasons? Even a table explaining how many were rejected for various reasons would be really helpful.
Response: We thank the reviewer for this comment. We have added an explanation of why studies were excluded during the screening process (Ln 346-350). We now write: “Throughout the screening, studies were excluded because either they did not deal with learning, did not examine the learning of L2 vocabulary, or did not compare gesture enactment with gesture observation (these studies compared gesture enactment with control conditions that required viewing pictures or written text).”
5. Results: I really like the table given very explicit details about the studies that were conducted. It makes it very clear to compare the 7 studies. I also really liked how the authors determined the level of bias and the figure that supported their argument. I think this study makes it clear that the question about whether gesture enactment or observation has not been adequately examined in the past, but that a pattern still exists. I wonder if in the discussion or conclusion that could be more adequately addressed -- that one of the most important findings of your study is that more rigor is needed to disentangle this question.
Response: We thank the reviewer for highlighting this point. We have added a paragraph emphasizing the need for more rigor in future research (Ln 615-625). We write: “A pivotal takeaway from our analysis underscores the need for not only more data that compares learning while viewing and enacting gestures, but also more rigorous research methodologies. The methodological limitations of the studies analyzed here impedes the formation of a clear consensus and calls into question the reliability of existing data. To enhance the quality of future research, it is vital that studies ensure comprehensive randomized assignment of participants to learning conditions or the use of within-group control conditions, utilize blinding techniques, and test large sample sizes when feasible [94, 95]. Studies could consider implementing peer reviews at the design stage or pre-registering study protocols, pursuing replication to validate findings, and involving multiple research centers to increase sample sizes. To truly understand the effects of gesture-based learning, it will be essential to uphold stringent research practices.”
Reviewer 2 Report
Comments and Suggestions for Authors
The paper presents a meta-analysis of studies concerning the exploitation of gesture as enhancer of vocabulary learning in L2, and in particular tries to assess if gesture enactment is more effective than gesture viewing in helping vocabulary learning, concluding with a negative answer to the question.
The topic is quite interesting, also because, besides its didactic implications, it tackles important theoretical issues concerning the representation of words and concepts in memory, bearing on embodied cognition and embodied learning theories; and the various models to describe them are reported in detail by the Authors.
The selection of the works to be included in the meta-analysis is very accurate and exploits quite sophisticated tools. The selection criteria are therefore very strict, maybe even too strict, so that, somehow disappointingly, from a total of 1298 records, the meta-analysis is reduced to only seven studies.
But besides this, the Authors’ review seems not to go so critically in depth into some details of the reviewed studies. Given that the review is finally reduced to so few works, not only the details of their procedure might have been described but the theoretical implications of the results obtained from certain details as opposed to others should have been overviewed. A first way to go more in depth would be to include in Table 1 an additional column with the studies’ results. This might at least in part amend the lack of theoretical discussion.
Actually, sometimes the paper fails to discuss issues that are not so straightforward.
For example, a fact that is not evidenced at all is that in all experiments, except perhaps for Krönke 2013 (but it is not so clear from Table 1), enactment was always preceded by observation. Yet, one of the main results of the studies seems to be that enacting is not better than observing; but in fact one should say that “observing + enacting” is not better that “observing” only. So one should try to account for this counterintuitive result, try to explain why this is so.
Again, from the description of the studies, it is quite easy to understand why iconic gestures may be useful for word retrieval, but it is less so for “pitch gestures”. Not only are they not defined at length by the Authors, but it is not clear how can they be a prompt to words (and not only to their accent or rhythm), since they have no strictly semantic content.
One more interesting case, quoted from the reviewed papers, but not discussed by the review Authors, is one of the “meaningless” gestures. Here one should predict that enacting meaningful ones is less effective than enacting meaningful ones. But the reader is left with this curiosity concerning the paper results. Another case in which a “result” column for Table 1 might be of help.
Author Response
We thank the Reviewer for their positive and constructive comments that have helped us to improve the manuscript. Please find a point-by-point reply to the comments below; reviewer comments are italicized and author replies are in normal font. Changes in the manuscript are highlighted with red font.
1. The selection of the works to be included in the meta-analysis is very accurate and exploits quite sophisticated tools. The selection criteria are therefore very strict, maybe even too strict, so that, somehow disappointingly, from a total of 1298 records, the meta-analysis is reduced to only seven studies.
Response: We thank the reviewer for this comment. We are currently starting a line of research in our department that compares the effects of enacting versus observing different types of gestures on L2 vocabulary learning. The strict selection criteria used in our meta-analysis were derived from this specific research question, which significantly narrowed down the studies that could be included. We agree that the pool of literature that has addressed this specific question is unfortunately limited. However, we hope that this highlights to the community the need for further clarifying studies, larger datasets, and more rigorous research that addresses this particular question.
2. But besides this, the Authors’ review seems not to go so critically in depth into some details of the reviewed studies. Given that the review is finally reduced to so few works, not only the details of their procedure might have been described but the theoretical implications of the results obtained from certain details as opposed to others should have been overviewed. A first way to go more in depth would be to include in Table 1 an additional column with the studies’ results. This might at least in part amend the lack of theoretical discussion.
Response: We appreciate this comment and thank the reviewer for suggesting this. To address the theoretical implications of the results obtained from certain details as opposed to others, we ran sub-group analyses (shown in Table 2), which cluster study results based on the studies’ methodological features. These sub-group analyses were followed up by moderation analyses, which test for effects of methodological features such as the number and type of words that participants were asked to learn. According to the guidelines of the Cochrane Handbook for Systematic Review of Interventions, we intentionally did not include individual study results in Table 1, as this could lead to a bias in readers’ interpretation of cross-study results. To avoid these kinds of biases, meta-analyses pool results from several studies, in this case providing the effect size of the comparison between gesture enactment and observation across studies. By weighing the results from each study, the meta-analysis procedure limits the bias related to considering the results of a single study over other studies.
3. Actually, sometimes the paper fails to discuss issues that are not so straightforward. For example, a fact that is not evidenced at all is that in all experiments, except perhaps for Krönke 2013 (but it is not so clear from Table 1), enactment was always preceded by observation. Yet, one of the main results of the studies seems to be that enacting is not better than observing; but in fact one should say that “observing + enacting” is not better that “observing” only. So one should try to account for this counterintuitive result, try to explain why this is so.
Response: We thank the reviewer for this thoughtful comment. We have added text to the Results section to make this point more explicit (Ln 361; 418-420). We also now further discuss this counterintuitive result in the Discussion section (Ln 541-554). We write: “In the studies assessed here, gesture enactment was accompanied by the observation of a model performing the same gestures. This means that observation was consistent across both ‘enactment’ and ‘observation’ scenarios. The absence of a distinct advantage when enacting gestures aligns with the notion that observation alone can give rise to sensorimotor simulations that are just as beneficial as overt action. Further, the current analysis does not indicate that combining enactment with observation negatively impacts L2 learning relative to observation alone, as would be predicted by theories emphasizing dual task interference effects in activities that involve physical movements [91]. Therefore, while enactment does not appear to provide an additional advantage relative to mere observation based on the current limited set of data, it does not detract from the learning experience either. This neutral effect of combined enactment and observation could suggest that, when gestures are incorporated into L2 learning, the primary driver of the learning benefit might be the observation and in-ternal simulation, rather than the physical enactment.”
4. Again, from the description of the studies, it is quite easy to understand why iconic gestures may be useful for word retrieval, but it is less so for “pitch gestures”. Not only are they not defined at length by the Authors, but it is not clear how can they be a prompt to words (and not only to their accent or rhythm), since they have no strictly semantic content.
Response: We thank the reviewer for highlighting this. We have added a section in the Introduction that defines pitch gestures and clarifies how they may be beneficial for vocabulary learning (Ln 84-95). We now write: “Interestingly, gestures need not always relate directly to word meanings to aid L2 learning. For instance, beat gestures—rhythmic hand or arm movements that emphasize specific words or mark rhythm without conveying specific meaning—have been found to benefit semantic learning [35]. Similarly, pitch gestures, which mirror speech intonation or melody, aid learners in discerning lexical tones and remembering the meanings of L2 words that contain those tones [14]. These findings hint at a relationship between auditory perceptual learning and semantic acquisition in which enhanced recognition of L2 perceptual patterns corresponds to improved L2 vocabulary learning. This is particularly evident in the case of non-tone-language speakers’ learning of tone languages, in which a change in pitch contour can alter word meaning. In such contexts, semantic proficiency has been shown to depend on the ability of a learner to differentiate pitch contours of lexical tones [36]”
5. One more interesting case, quoted from the reviewed papers, but not discussed by the review Authors, is one of the “meaningless” gestures. Here one should predict that enacting meaningful ones is less effective than enacting meaningful ones. But the reader is left with this curiosity concerning the paper results. Another case in which a “result” column for Table 1 might be of help.
Response: We appreciate this helpful comment, and we agree that it would be very interesting to pool the results of studies comparing observation with enactment of meaningful and meaningless gestures. However, only two of the included studies made such comparison. Looking at these two studies independently, García-Gámez et al. (2021) found statistical differences between these conditions, while Krönke et al. (2013) did not. We agree, however, that this aspect should be mentioned in the manuscript. Rather than speculating on the potential direction of the results, we discuss this as an open avenue for future research (Ln 626-640). We write: “Two additional variables to explore in future work are the degree of congruency between gestures that are performed or observed during L2 learning and the meanings of L2 words, as well as the role of self-generated (i.e., self-created) gestures in L2 learning. The enactment of gestures that are incongruent with word meanings and the enactment of meaningless gestures—movements that do not carry inherent symbolic or semantic value—may be detrimental to L2 learning rather than facilitatory [15]. Manipulations of between-modality congruency may help in quantifying the role of physical activity or engagement in learning, independent of a gesture’s semantic content. On a related note, the role of self-generated gestures in L2 learning remains largely unexplored. Self-generated gestures allow learners to create their own physical representations of L2 vocabulary, potentially leading to a deeper, more personalized understanding. Evidence from other domains suggests that self-generated content can enhance memory retention and understanding [96, 97]. It is possible that when learners create their own gestures, they form stronger or more memorable representations of the vocabulary, which could lead to greater benefits of enactment relative to observation alone.”